# Liquid Biopsy in Lung Cancer: Biomarkers for the Management of Recurrence and Metastasis

**DOI:** 10.3390/ijms24108894

**Published:** 2023-05-17

**Authors:** Vanessa G. P. Souza, Aisling Forder, Liam J. Brockley, Michelle E. Pewarchuk, Nikita Telkar, Rachel Paes de Araújo, Jessica Trejo, Katya Benard, Ana Laura Seneda, Iael W. Minutentag, Melis Erkan, Greg L. Stewart, Erica N. Hasimoto, Cathie Garnis, Wan L. Lam, Victor D. Martinez, Patricia P. Reis

**Affiliations:** 1British Columbia Cancer Research Institute, Vancouver, BC V5Z 1L3, Canada; aforder@bccrc.ca (A.F.); lbrockley@bccrc.ca (L.J.B.); mpewarchuk@bccrc.ca (M.E.P.); ntelkar@bccrc.ca (N.T.); jtrejo@bccrc.ca (J.T.); kbenard@bccrc.ca (K.B.); gstewart@bccrc.ca (G.L.S.); cgarnis@bccrc.ca (C.G.); wanlam@bccrc.ca (W.L.L.); 2Molecular Oncology Laboratory, Experimental Research Unit, School of Medicine, São Paulo State University (UNESP), Botucatu, SP 18618-687, Brazil; rachel.paes@unesp.br (R.P.d.A.); ana.seneda@unesp.br (A.L.S.); iael.weissberg@unesp.br (I.W.M.); patricia.reis@unesp.br (P.P.R.); 3British Columbia Children’s Hospital Research Institute, Vancouver, BC V5Z 4H4, Canada; 4Department of Pathology and Laboratory Medicine, IWK Health Centre, Halifax, NS B3K 6R8, Canada; merkan@dal.ca; 5Department of Pathology, Faculty of Medicine, Dalhousie University, Halifax, NS B3K 6R8, Canada; 6Beatrice Hunter Cancer Research Institute, Halifax, NS B3H 4R2, Canada; 7Department of Surgery and Orthopedics, Faculty of Medicine, São Paulo State University (UNESP), Botucatu, SP 18618-687, Brazil; erica.hasimoto@unesp.br; 8Division of Otolaryngology, Department of Surgery, University of British Columbia, Vancouver, BC V5Z 1M9, Canada

**Keywords:** CTCs, ctDNA, liquid biopsy, metastasis, recurrence, lung cancer

## Abstract

Liquid biopsies have emerged as a promising tool for the detection of metastases as well as local and regional recurrence in lung cancer. Liquid biopsy tests involve analyzing a patient’s blood, urine, or other body fluids for the detection of biomarkers, including circulating tumor cells or tumor-derived DNA/RNA that have been shed into the bloodstream. Studies have shown that liquid biopsies can detect lung cancer metastases with high accuracy and sensitivity, even before they are visible on imaging scans. Such tests are valuable for early intervention and personalized treatment, aiming to improve patient outcomes. Liquid biopsies are also minimally invasive compared to traditional tissue biopsies, which require the removal of a sample of the tumor for further analysis. This makes liquid biopsies a more convenient and less risky option for patients, particularly those who are not good candidates for invasive procedures due to other medical conditions. While liquid biopsies for lung cancer metastases and relapse are still being developed and validated, they hold great promise for improving the detection and treatment of this deadly disease. Herein, we summarize available and novel approaches to liquid biopsy tests for lung cancer metastases and recurrence detection and describe their applications in clinical practice.

## 1. Introduction

Lung cancer is the leading cause of cancer-related death worldwide, with an estimated 1.8 million deaths in 2020 [1]. Despite significant progress in understanding and treating lung cancer, the 5-year relative survival rate remains relatively low (25.4% from 2013 to 2019), as reported by the Surveillance, Epidemiology, and End Results (SEER) Program of the National Cancer Institute [2]. Lung cancer has a high mortality rate, mainly due to diagnosis occurring at later stages, as the time from onset to symptom presentation is largely variable, attributed to the slow-growing rate of lung tumors [3]. As a result, many lung cancer patients are diagnosed at advanced stages, when the disease has become locally invasive and/or has spread to distant organs, making it difficult to administer treatment, resulting in a poor prognosis [4]. 

Metastasis is a complex process by which cancer cells spread from the primary tumor to distant sites in the body, such as the bones, liver, or brain [5,6,7]. Cancer-related deaths are overwhelmingly the result of this disease progression (>90%), where metastasis leads to the impairment of vital organ functions [8,9]. The likelihood of metastasis depends on the histological type of cancer and the patient’s overall health. Approximately 30–40% of non-small-cell lung cancer (NSCLC) patients have metastatic disease at the time of diagnosis [10,11]. For small-cell lung cancer (SCLC), the incidence of metastasis at diagnosis is higher (~60%) due to the aggressive nature of this cancer type [12]. 

Recurrence occurs when cancer cells reappear after remission [13]. Recurrence can be especially challenging to manage, as the cancer may have become increasingly resistant to treatment over time. Several traits influence the risk of recurrence, including the type and stage of cancer and the effectiveness of the primary treatment [14,15,16]. Approximately 30–55% of patients with NSCLC develop recurrence, which proves fatal, despite multiple curative resections [14,17,18]. 

Treatment options for advanced lung cancer are limited, and most patients are not candidates for curative treatment. In these cases, molecular testing plays a crucial role in identifying actionable alterations that can guide the selection of targeted therapies and immunotherapies; the use of targeted therapies, such as *EGFR* and *ALK* inhibitors, has proven beneficial for improved treatment outcomes and reduced toxicity when compared to traditional chemotherapy [19,20,21,22,23]. Similarly, immunotherapies, such as *PD-1* and *PD-L1* inhibitors, have demonstrated significant clinical benefits in advanced lung cancer patients [24,25,26]. However, the challenge lies in obtaining sufficient tumor tissue for molecular testing in patients with advanced disease stages, as they may not be eligible for repeated biopsies, depending on the patient performance status, tumor location, or extent of disease [27]. Furthermore, repeat biopsies carry their own risk of complications [28]. In these cases, fine needle biopsies or cytology specimens are the only potential sources of tissue for molecular testing, but the limited amount of tumor material obtained from these can sometimes be insufficient for accurate testing. 

Liquid biopsy is a minimally invasive procedure that involves the analysis of tumor-derived materials such as circulating tumor cells (CTCs), cell-free DNA (cfDNA), and extracellular vesicles (EVs) that can be processed from various body fluids including blood, urine, and pleural effusion [29]. Liquid biopsy is more globally applicable to aid in tumor screening, tumor staging, the characterization of intratumoral heterogeneity, the monitoring of tumor progression, and the therapy response for individualized treatment [30]. The application of liquid biopsies has proven to be a significant asset in lung cancer’s early detection and treatment [31]. In addition, liquid biopsies have the potential to significantly improve the management of patients with advanced lung cancer, by providing non-invasive and real-time monitoring of disease progression, the treatment response, and recurrence. A recent meta-analysis showed that ctDNA detection can be more sensitive to predicted tumor recurrence in lung cancer patients than CTCs, although both have potential results [32]. It has also been indicated that liquid biopsies are useful in guiding adjuvant and/or neoadjuvant treatment, predicting prognosis, and detecting molecular changes related to tumor resistance [29,33,34,35]. Furthermore, liquid biopsies show promise as an alternative by enabling a broad range of cytological and molecular evaluation methods through minimally invasive procedures [36,37].

As technology continues to evolve, liquid biopsies are slowly becoming indispensable for the clinical management of advanced lung cancer. Although previous studies have investigated liquid biopsies in lung cancer [38,39,40,41,42], our review offers novel insights by specifically highlighting their potential applications for the management of metastasis and recurrence in NSCLC. Here, we comprehensively review relevant data, contributing to the literature by thoroughly elucidating the role of liquid biopsies with a focus on clinical applications for the better management of metastatic and recurrent NSCLC, which are the most serious and poorly managed clinical complications of lung cancer leading to treatment failure and patient death.

## 2. Current Liquid Biopsy Samples and Biomarkers for Late-Stage Lung Cancer

Liquid biopsy for advanced-stage lung cancer typically involves the analysis of samples such as peripheral blood, urine, and cerebrospinal fluid (CSF), which contain various biomarkers, including ctDNA, CTCs, EVs, and microRNAs. Here, we provide an overview of the current state of knowledge on the use of liquid biopsies in detecting recurrence and metastasis in lung cancer, including a description of the various biomarkers and technologies used in liquid biopsies. 

### 2.1. Sample Subtypes of Liquid Biopsies Used in Advanced-Stage Lung Cancer

The examination of body fluids helps to further determine the characteristics of lung cancer by identifying the presence of cancer cells or substrates secreted by them [43]. The levels of these molecules can vary according to different conditions, such as stress factors, tumor location, status, metastasis extent, and genome instability [44]. Therefore, these indicators serve as biomarkers, allowing detection, assessment, and diagnosis. 

In the advanced stages of NSCLC, the initial clinical application for liquid biopsy was the detection of sensitizing *EGFR* mutations, which may be associated with an enhanced tumor response. In addition, the evaluation of acquired resistance in samples such as peripheral blood is a supplementary tool to tissue analysis used in clinical examination [33]. Currently, with the implementation of broad-based platforms such as next-generation sequencing (NGS), the additional genotyping of oncogene drivers supports the utility of plasma ctDNA, as reviewed elsewhere [45,46,47,48]; in fact, the absence of detectable alterations in plasma has been associated with a lower tumor burden [49]. However, the use of plasma is not sufficient for patients with metastasis exclusive to the brain or leptomeninges, and, for these cases, the use of CSF arises as an alternative source of ctDNA [50].

Table 1 lists examples of biomarkers that can be detected in body fluids, such as plasma, serum, sputum, bronchoalveolar lavage (BAL), pleural effusion, urine, and cerebrospinal fluid (CSF), and their applications in cancer diagnosis, prognosis, and treatment. Examples of biomarkers include ctDNA, miRNA, gene mutations, and protein signatures. Such biomarkers can help to determine resistance mechanisms, predict clinical outcomes, and monitor treatment efficacy. 

### 2.2. Biomarker Subtypes of Liquid Biopsies Used in Advanced-Stage Lung Cancer

Several types of biomarkers have been reported in the context of liquid biopsy for advanced-stage lung cancer. These include CTCs, ctDNA, cell-free RNA (cfRNA), EVs, DNA methylation markers, and other circulating elements.

#### 2.2.1. Extracellular Vesicles (EVs)

EVs are nano-sized lipid-bound particles that are secreted by cells into the extracellular space [73,74,75]. There are three main subtypes of EVs: microvesicles (MVs), exosomes, and apoptotic bodies, which can be differentiated based on their biogenesis, release pathways, size, content, and function [73]. EVs have emerged as promising biomarkers for liquid biopsy in lung cancer due to their ability to carry various types of molecular cargo, including proteins, lipids, DNA, RNA, and other important cellular components [76,77,78,79]. EVs are released by tumor cells and can be detected in blood, saliva, urine, and other body fluids. They have been shown to play a role in cell-to-cell communication, promoting tumor growth, angiogenesis, immune evasion, and metastasis in lung and other cancers [77,80,81]. Several studies have demonstrated the potential of EVs as liquid biopsy biomarkers for lung cancer. For example, EV-associated mutations in oncogenic driver genes, such as *EGFR* and *ALK*, have been detected in EV samples from lung cancer patients, and their presence has been correlated with the tumor mutational status and treatment response [69,82,82,83,84,85]. EV-based liquid biopsy has also been shown to be able to detect minimal residual disease (MRD) and monitor disease recurrence in lung cancer patients after surgery or other treatments, providing valuable information for personalized treatment decisions and surveillance [86,87].

In addition, studies have demonstrated that cancer cells release high levels of EVs, which can facilitate multiple steps in the metastatic cascade. EVs can promote tumor cell proliferation, migration, and invasion, and can also modulate the tumor microenvironment, promoting angiogenesis and immune evasion [88,89,90,91]. Moreover, EVs released by cancer cells can promote the formation of a pre-metastatic niche in distant organs, preparing the microenvironment for the arrival of cancer cells and facilitating their growth and colonization [90,92,93]. EVs can also facilitate the spread of cancer by supporting the survival and growth of CTCs in the bloodstream. By carrying bioactive molecules, EVs can protect CTCs from immune surveillance and apoptosis, and can also promote the formation of CTC clusters or aggregates with platelets and leukocytes, which can facilitate their survival in the bloodstream [94,95].

#### 2.2.2. Circulating Tumor DNA (ctDNA)

ctDNA is a subset of cfDNA (circulating free DNA) that originates from tumor cells [44,96]. cfDNA refers to the fragmented DNA that is found in the bloodstream of an individual. The levels of total cfDNA are often higher in individuals with cancer compared to those without cancer, and this is thought to be due in part to the presence of ctDNA [97]. The amount of ctDNA released into the bloodstream can vary between different types of cancer, as well as between individual patients with the same type of cancer [98]. 

ctDNA levels are generally measured from plasma, as serum has a high likelihood of being contaminated by the release of genomic DNA from white blood cells during clotting [99]. ctDNA levels in plasma are generally low, approximately 5–10 ng/mL [100], and so the method of detection used is incredibly important. Droplet digital polymerase chain reaction (ddPCR) and NGS have greatly improved ctDNA’s detection sensitivity in recent years [101,102], and there are a large number of metrics that can be used for ctDNA analysis, including ctDNA concentration/levels, the quantitation of tumor-specific mutations in the ctDNA, and differing methylation patterns indicating epigenetic changes [103].

ctDNA-based biomarkers have been approved for clinical use in NSCLC, including *EGFR* mutation detection and NGS screening of a mutation panel, which includes *ALK*, *EGFR*, and *KRAS* [104]. However, this does not reflect the variety of recent advances in the field. ctDNA has many applications in advanced/metastatic lung cancer [105], including predicting the treatment response and survival for patients under treatment with chemotherapy regimens [106], targeted therapies [107,108], immune checkpoint inhibitors [109], and chemo-immunotherapy [110]. A recent study showed that longitudinal dynamic changes in ctDNA metrics using a machine learning model are associated with patient survival in metastatic NSCLC, with the potential utility of ctDNA for predicting treatment outcomes early in clinical trials, as ctDNA outperformed radiographic endpoints [105]. Similarly, serial ctDNA analysis has been shown to identify recurrence and distant metastasis earlier than conventional radiologic imaging [111]. Furthermore, the detection of ctDNA after the treatment of NSCLC could predict early recurrence after treatment [112], and clonal ctDNA mutations were shown to be a marker for MRD in NSCLC after surgical resection [113]. An increased number of variants have been linked to prognosis in advanced NSCLC and may predict lymph node metastasis even at the preoperative stage [114]. Therefore, ctDNA analysis shows great promise for liquid biopsy applications in lung cancer, but the challenge remains in translating findings into routine clinical practice.

#### 2.2.3. Cell-Free RNAs (cfRNAs)

cfRNAs are detected in the bloodstream and other body fluids. Similarly to ctDNAs, a subset of these may be released from tumor cells [115,116]. There are several classes of RNA that can be found in circulation, including messenger RNAs (mRNAs) and non-coding RNAs (ncRNAs). Quantification is usually undertaken using PCR-based assays such as reverse-transcription quantitative PCR (RT-qPCR) and ddPCR, and advancements in NGS methods have been invaluable for biomarker discovery [117,118,119,120].

Within mRNAs, PD-L1 mRNA levels have been useful in predicting sensitivity to anti-PD-L1 therapy [121]. Beyond the therapy response, measuring the levels of mRNAs from saliva in combination with the presence of CTCs has been shown to discern NSCLC patients from healthy controls [122]. However, due to the presence of ribonucleases in the blood, mRNAs are generally fragmented in circulation and thus give low-quality reads when analyzed by sequencing [116]. 

ncRNA species have garnered significant interest due to their association with exosomes and protein complexes, which can protect them from degradation [115,116,118,123]. In particular, microRNAs (miRNAs) are of great interest as biomarkers and are the most abundant cfRNA species [118]. There have been several examples of miRNAs that show an association with metastases in NSCLC, such as miR-422-a with lymph node metastasis [124], and miR-483-5p and miR-342-5p with leptomeningeal metastasis [125]. The expression of miRNAs can, however, be highly variable between patients and requires validation [126,127].

Beyond miRNAs, ncRNAs such as long non-coding RNAs (lncRNAs), tRNA-derived fragments (tRFs), circular RNAs (circRNAs), and PIWI-interacting RNAs (piRNAs) can also be detected in liquid biopsy samples [118,123]. In lung cancer, the lncRNA metastasis-associated lung adenocarcinoma 1 (*MALAT-1*) is often differentially expressed in tumor samples compared to normal ones and may promote brain metastasis, which has led to multiple studies looking into its utility as a biomarker for NSCLC [128,129,130]. Although lncRNAs are one of the lowest-abundance cfRNAs in circulation, *MALAT-1* is detectable in serum and may complement biomarker panels to improve NSCLC patient diagnosis and prognosis [130].

While much more is still to be learned about the utility of cfRNAs in advanced-stage disease, there has been some success with the development of the blood-based neuroendocrine neoplasms test (NETest), which has been used in the monitoring of gastrointestinal neuroendocrine tumors by profiling the gene expression of 51 mRNAs from peripheral blood [131,132]. However, there have not been any clinical tests for lung cancer cfRNAs developed yet.

#### 2.2.4. DNA Methylation Markers

DNA methylation, which includes the dynamic addition of a methyl group to a cytosine nucleotide, converting the cytosine into a 5-methylcytosine (5mC), can greatly influence gene and downstream protein expression. Aberrant hypermethylation is a common finding within tumor tissue when compared to generally normal and healthy tissue. Global blood-based methylation biomarkers obtained from liquid biopsy material (cfDNA, ctDNA) have been difficult to determine; however, methylation profiles for lung cancer subtypes have shown robust differences between diseased and control states [133,134,135,136,137], and differences can even be detected between circulating tumor material and direct tumor tissue [138]. These profiles show remarkable differences even in early cancer stages, with certain methylation markers specific to particular cancers, whereas some are detectable pan-cancer [139,140]. Some markers have notable specificity to lung cancer metastasis [141]; however, the detection of these metastasis markers, and furthermore of recurrence, is a relatively novel and ongoing area of research [141,142,143].

The quality of methylation profiling is significantly influenced by the method of detection and analysis. While bisulfite conversion is still considered the gold standard, its complex processing can result in higher error rates. The Infinium HumanMethylation450 BeadChip array and the Human MethylationEPIC BeadChip are currently the most widely used methylation profiling platforms.

Methylation signatures are powerful cancer screening tools, where methylation scores have been shown to not only accurately classify cancer-derived samples from non-cancerous, but also predict cancer subtypes with high accuracy [144]. Especially in instances of a high cancer predisposition and the subsequent prediction of cancer onset, methylation profiling can provide an inexpensive yet efficient method for cancer management.

## 3. Monitoring Minimal Residual Disease (MRD)

After curative treatment, clinical follow-up must include the evaluation of MRD, which refers to the small number of cancer cells that may remain in the patient’s body after treatment, even if they cannot be detected by standard imaging tests [145]. These residual cells have the potential to grow and cause recurrence, making MRD an important tool for monitoring treatment responses and predicting outcomes [112].

While genetic profiling has greatly increased therapy success, fatal disease recurs in 30–70% of resected cases [32,146]. Lung cancer patients generally respond well to initial treatment; however, they develop resistance approximately two years later due to acquired or de novo molecular alterations, which might have been present in lower frequencies in the cells before the treatment [147,148,149,150]. Once the treatment kills the sensitive target cells, pre-existing resistant subclones are released to proliferate, inducing the emergence of a tumor population resistant to the applied therapy, i.e., MRD [147].

MRD detection is challenging, particularly in the context of lung cancer, because these tumors tend to develop chromosomal instability later in their natural history [151]. Clonal evolution and tumor heterogeneity are important factors to consider in monitoring MRD, as they can affect the accuracy of detection and the risk of recurrence. Throughout the natural history of the disease, the genomes of cancer cells accumulate changes, and they mutate and evolve in response to treatment and other environmental factors. This process, called clonal evolution, can lead to the emergence of new cancer cell subclones, leading to an increase in tumor heterogeneity, which can contribute towards treatment resistance and recurrence [147,151,152].

### 3.1. Clonal Evolution and Tumor Heterogeneity within MRD

As cancer grows by clonal evolution, dynamic molecular changes are acquired, generating subclonal cell variation, which can be defined as intratumoral heterogeneity (ITH). These changes include genetic, epigenetic, and gene expression alterations and may impact the antitumoral immune response. The advances in NGS and single-cell sequencing have enabled better knowledge of such alterations and their clinical implications, predicting the risk of metastasis, treatment response, and resistance [147,151]. Unfortunately, treatment resistance is common and can arise through various mechanisms, including point mutations, gene fusions, and cellular plasticity, giving rise to unpredictable phenotypes. As a result, employing multiple treatment modalities can enhance the efficacy, but combination therapies may also escalate treatment-related toxicity [151,152]. Mechanisms that cause lung adenocarcinomas with mutant *EGFR* to become resistant to osimertinib have been proposed. Tumors can show two or more resistance mechanisms, such as focal copy-number amplifications in *MET*, *KRAS*, and *PD-L1*, gene fusions of *ALK*, *MKRN1-BRA*, and *ESR1-AKAP12*, as well as neuroendocrine differentiation [153]. Therefore, combination therapy strategies have been strongly indicated in these cases [154].

In the context of ITH, liquid biopsies offer a range of potential applications, including early detection, the detection of MRD, the guidance of treatment decisions, the investigation of treatment efficacy and resistance, and real-time disease monitoring [40,155]. 

### 3.2. Minimal Residual Disease (MRD) and the Role of Liquid Biopsy in Detecting Lung Cancer Recurrence

Currently, the genetic profiling of solid tumors and detection of MRD in patients with lung cancer who have undergone curative treatment are primarily achieved using surgical or biopsy specimens. However, these methods have several limitations, such as invasiveness, making routine implementation challenging. Additionally, they provide only a temporally limited snapshot of the tumor and may fail to capture disease heterogeneity [156]. In contrast, liquid biopsy is a non-invasive method for longitudinally assessing MRD at any time point post-surgery [157].

Studies have shown that liquid biopsies are able to detect MRD well before the detection of recurrence by standard clinical methods [157,158,159]. A systematic review of 13 studies using ctDNA to detect MRD in NSCLC patients found that detection was achieved an average of 5.5 months prior to detection by radiography or other clinical methods [158]. One of these studies investigated perioperative ctDNA detection as a marker of MRD in 330 patients with stage I-III NSCLC with a custom panel of 769 genes. Negative perioperative ctDNA detection was more relevant in predicting recurrence-free survival than any other factor considered, including TNM staging of the extent of cancer progression [160]. A recent ten-year study of 13 breast cancer patients that used multiple biomarker detection methods derived from both CTCs and ctDNA in liquid biopsy samples found that these combined tests could uncover MRD at least four years prior to the emergence of clinically detectable metastases [161]. Another study evaluated the neoadjuvant immunotherapy efficacy (NAT) in resected NSCLC and assessed the risks for recurrence using ctDNA. During immunotherapy, ctDNA detection was highly concordant with the pathologic response, with overall accuracy of 91.67%. After three months of surgery, ctDNA was able to predict the patient’s recurrence with 83% sensitivity and 90% specificity. The recurrence prediction using ctDNA was capable of anticipating the radiographic recurrence, with a median time of 6.83 months [162].

Liquid biopsy tests for MRD can be divided into tumor-informed (requires knowledge of the tumor’s genetic profile beforehand by examining a sample of the tumor or analyzing preoperative cell-free DNA) and tumor-uninformed (intended to identify the existence of MRD without any prior knowledge of the specific molecular alterations that may be present in the tumor of an individual patient) assays [163]. Assays that use the tumor-informed approach, such as RaDaR [164], Signatera [165], and the recently tested HIFI platform [47], are personalized using the baseline genotype of the patient’s primary tumor or preoperative cfDNA and require longitudinal follow-up to probe for emergent mutations. This strategy has higher sensitivity and specificity than the tumor-uninformed approach but can have a delayed turnaround time. The tumor-uninformed approach includes assays such as DELFI [166]. With this approach, the same assay is used for each patient, which leads to a shorter turnaround but diminished specificity and sensitivity. Trials of liquid biopsy assays for MRD detection in NSCLC patients are currently underway [163].

## 4. Metastasis

Metastasis involves the successful completion of the metastatic cascade (Figure 1). During the metastatic cascade, biomarkers released directly from cancer cells or associated with cancer cells can be used as potential biomarkers in liquid biopsy. These biomarkers, including CTCs, ctDNA, exosomes, and specific proteins, hold potential as indicators of cancer metastasis and can be detected and analyzed in liquid biopsy samples. They offer a non-invasive and dynamic approach to assessing the presence and characteristics of metastatic tumors.

### 4.1. The Metastatic Cascade and Associated Biomarkers

Metastasis is a complex process by which cancer cells spread from the primary tumor to distant sites in the body [167]. In lung cancer, metastasis commonly occurs through the bloodstream or lymphatic system, leading to the development of secondary tumors in other organs, such as the brain, liver, and bones [5]. Chambers et al. [168] proposed a model called the “metastatic cascade” to describe the process of cancer dissemination. The metastatic cascade involves several steps, including the detachment of cells from the primary tumor, invasion of surrounding tissues (intravasation), spread throughout the circulatory system or lymphatic vessels (circulation), extravasation into distant tissues, formation of micrometastases, and establishment of a secondary tumor (colonization) [7,167,169,170,171].

During the metastatic cascade, a series of complex events result in the release of biomarkers into the bloodstream [172]. These biomarkers play a significant role in the biological processes involved in metastasis. For example, EVs released by cancer cells can have a multifaceted influence on endothelial cells (ECs). The interaction between tumor-derived EVs and ECs can promote tumor angiogenesis, trigger endothelial–mesenchymal transition (EMT), and disrupt the endothelial vascular barrier [91,173,174,175]. This disruption allows CTCs to cross the endothelial barrier and enter the bloodstream, facilitating the process of metastasis. Furthermore, EVs can indirectly induce tumor angiogenesis by promoting the phenotype switching of different cell types into cancer-associated fibroblasts, activating tumor-associated ECs and platelets, and remodeling the extracellular matrix [91,173]. Additionally, tumor-derived EVs released by tumor-educated platelets (TEPs) may enhance the permeability of endothelial cells by secreting adenosine triphosphate (ATP), which can modulate the dynamics of endothelial cell junctions and facilitate CTCs’ extravasation [176].

Other examples of biomarkers include platelets and neutrophils, which help CTCs to evade the immune system and survive in the bloodstream by forming clusters or aggregates with CTCs [177,178,179]. EVs contain integrins, such as αVβ5 for liver metastasis or α6β4 and α6β1 for lung metastasis, that are responsible for guiding cancer cells to specific organs, contributing to the formation of the pre-metastatic niche and influencing the site of metastatic colonization [180,181]. Proteins such as matrix metalloproteinases (MMPs) [182,183,184,185] and vascular endothelial growth factor (VEGF) facilitate angiogenesis, tumor cell invasion, immune surveillance escape, and the preparation of the pre-metastatic niche [186,187,188,189].

### 4.2. Applications of Liquid Biopsy in Metastatic Lung Cancer

#### 4.2.1. Liquid Biopsy for the Early Detection of Lung Cancer Metastasis

One major role for liquid biopsy in managing lung cancer metastasis is in the context of leptomeningeal metastasis (LM), which occurs in approximately 3–5% of NSCLC and is more common in *EGFR*-mutant NSCLC [190,191]. Because this metastatic site is not accessible for tissue biopsy, the current gold standard for diagnosis is the presence of neoplastic cells in the CSF [192]. Several studies have reported that the CSF is a more suitable sample for liquid biopsy than serum or plasma in the context of LM and other central nervous system metastases. ctDNA in the CSF of NSCLC patients with central nervous system metastasis has a more complete genomic alteration profile than ctDNA in matched plasma samples [193]. ctDNA in the CSF of NSCLC with LM was shown to be more comprehensive for profiling driver mutations than plasma, regardless of the presence or absence of an additional extracranial metastatic site [194]. Targeted NGS of cancer-relevant genes in the CSF of NSCLC patients with central nervous system metastasis revealed that CSF had higher sensitivity for identifying actionable driver mutations than plasma [195], and CSF was reported to be a more representative liquid biopsy sample type for LM in the context of EGFR-mutant NSCLC [191]. Another study reported that cfDNA in the CSF of NSCLC with LM showed a higher variant allele frequency than matched plasma, hypothesizing that the enrichment of ctDNA in the CSF may explain the high sensitivity of CSF compared to plasma in this context, and showed that a high level of genomic instability in cfDNA in the CSF was associated with shorter overall survival. Although CSF is the most commonly used liquid biopsy medium for NSCLC with LM, obtaining CSF is invasive as it requires the use of a lumbar puncture. A recent study reported that miR-483-5p and miR-342-5p, encapsulated in exosomes in the serum, may be involved in the LM of lung cancer and could offer a non-invasive alternative to the use of CSF for predicting LM in NSCLC. Additionally, miR-330-3p levels have been shown to be higher in NSCLC patients with brain metastasis [196]. 

Beyond LM and other central nervous system metastases, blood is generally used for liquid biopsy in lung cancer. For example, miR-422a in the plasma demonstrated diagnostic value for lymphatic metastasis [125]; exosomal miR-210-3p, miR-193a-3p, and miR-5100 in the plasma could discriminate between those with metastasis and those without [197]; and the downregulation of exosomal miR-574-5p and upregulation of exosomal miR-328-3p and miR-423-3p were observed in the plasma of lung cancer patients with bone metastasis versus those without [198]. The concentration of circulating free DNA in plasma from a lung cancer cohort was also associated with the stage of the lung cancer and the number of metastatic sites present [199], and the presence of CTCs in advanced NSCLC was associated with poor prognosis and the existence of distant metastatic sites [200]. A bioinformatic tool called ECLIPSE was recently developed and used to analyze the subclonal architecture in ctDNA from plasma samples from the TRACERx cohort, demonstrating that subclonal expansion in the primary tumor can be measured using ctDNA and is associated with metastatic potential [201]. ECLIPSE was developed specifically to handle the low ctDNA levels common in early-stage NSCLC so that liquid biopsy can be used to identify patients at higher risk for future metastases.

#### 4.2.2. Liquid Biopsy to Monitor Treatment Response and Identify Actionable Mutations in the Clinic

Liquid biopsies have also been shown to be effective in monitoring treatment responses in lung cancer patients, and the genetic profiles obtained from liquid biopsies tend to be consistent with those obtained from tissue biopsies. A study evaluated the concordance between liquid biopsies and tissue biopsies in detecting driver mutations in advanced non-small-cell lung cancer patients and reported a concordance rate of 85.7% [202]. Another study has reported a concordance rate of 70% to 80% between liquid biopsies and tissue biopsies in detecting EGFR mutations in non-small-cell lung cancer patients [203]. Changes in the levels of ctDNA during systemic treatments such as chemotherapy [106,204], targeted therapies [107,108], and immune checkpoint inhibitors [106,109] have been linked to treatment responses and survival times in patients with metastatic cancer. Recently, Assaf et al. showed that the use of ctDNA metrics over multiple time points can be beneficial for risk stratification and survival prediction in patients with metastatic NSCLC who receive chemo-immunotherapy combinations [105]. Such studies demonstrate the clinical applications of liquid biopsy tests for the prediction of the immunotherapy response in patients with advanced NSCLC. More recently, the surveillance of circulating biomarkers, mainly ctDNA and CTCs, has been suggested as a beneficial resource for clinicians when making therapeutic decisions during the immunotherapy of patients with advanced NSCLC [205]. A concrete example of using serial liquid biopsy to guide individualized treatment for a patient with advanced NSCLC that later progressed to bone metastasis has been reported in a recent case study. The patient was initially started on an *ALK* inhibitor after an *ALK* rearrangement was noted by fluorescence in situ hybridization (FISH) and switched over to a newly approved *KRAS*-p.G12C inhibitor after ctDNA analysis subsequent to disease progression revealed no *ALK* rearrangement but a *KRAS*-p.G12C mutation [206].

## 5. Nucleic-Acid-Based Methods for Detecting Lung Cancer in Liquid Biopsies

The most common approaches for nucleic-acid-based detection in liquid biopsy samples include multiplex-hybridization-based methods, PCR-based methods, and NGS. These approaches vary in sensitivity and in the variants that they are capable of detecting, as reviewed in our previous paper [45]. Table 2 displays the technical approaches based on their sensitivity, multiplexing capability, turnaround time (TAT), and limit of detection (LOD). Long-read sequencing can be used to analyze fragmentomic signatures of circulating DNA, and single-cell sequencing can be used to study cancer metastasis, evolution, and progression.

### 5.1. Multiplex-Hybridization-Based and PCR-Based Methods

Multiplex-hybridization-based and PCR-based assays are typically used to screen for variants in a predetermined gene panel. The nCounter technology is a multiplex-hybridization-based assay that has been evaluated for liquid biopsy applications in NSCLC, as this platform is capable of generating high-quality gene expression data from low-quality genomic material [201]. This platform has been shown to have similar performance to NGS [202]. A comparison between two PCR-based systems (ddPCR-BioRad and Qiagen QIAcuity Digital PCR) for the liquid biopsy analysis of cfDNA in a clinical study of NSCLC patients showed that solid digital PCR was generally more sensitive, and *KRAS* and *EGFR* mutations were detected with differing sensitivities [207]. 

### 5.2. NGS Methods

Given the large number of possible oncogene targets in advanced cancers, the wider genomic range of NGS is more applicable to novel biomarker discovery and can be comparable to hotspot mutation panels in sensitivity and cost [50,208]. Some NGS platforms for ctDNA analysis have been FDA-approved for use in clinical settings: the Guardant360 [209] and FoundationOne Liquid CDx [210] platforms can both be used to detect clinically actionable mutations in ctDNA for late-stage NSCLC cancer cases and have been recommended in lieu of smaller panels such as EGFR mutation PCR tests [50]. Targeted NGS panels such as these sequence a large but limited set of cancer-associated genes, and they have high sensitivity since they sequence genes of interest with high depth, but rare variants are not within the scope of analysis. Whole-genome or whole-exome sequencing can screen for all variants, but the tradeoff is lower sensitivity in comparison with targeted NGS due to the decreased sequencing depth, which can be prohibitive when there are low concentrations of ctDNA available for analysis [211]. 

**Table 2 ijms-24-08894-t002:** Capabilities of selected sequence-based detection methods that have been used to analyze liquid biopsy samples in the context of lung cancer. The limit of detection (LOD) for a genetic test is generally defined as the lowest mutant allele fraction (MAF) that can be detected with a given probability (usually 95%) and depends on the sample type, the sample concentration, and the variant being detected.

Detection Method	Multiplexing (Number of Markers)	Turnaround Time (TAT)	Sensitivity/Limit of Detection (LOD)
Multiplex-hybridization-based methods	nanoString nCounter: 800+ target genes	Dependent on panel [212]	Sensitivity of 95% and specificity of 82% [213] 0.02–2% MAF [214]
PCR-based methods	Small, predetermined gene panels	~2–3 days depending on panel [72]	qPCR: >10% MAF [215] dPCR: ~0.01% MAF [216]
NGS-based methods	WES: entire exome. WGS: entire genome. Targeted panels: large number of genes (e.g., TSO500 uses 500-gene panel [217])	~13 days depending on panel [72,218]	<1% MAF, can be <0.1% with specialized methods [211]

Abbreviations: dPCR: digital polymerase chain reaction; qPCR: quantitative polymerase chain reaction; MAF: mutant allele fraction; WES: whole-exome sequencing; WGS: whole-genome sequencing.

### 5.3. Fragmentomics and Long-Read Sequencing

Fragmentomics is the analysis of the non-random fragmentation patterns of cfDNA [219], and these fragments have been observed to be promising biomarkers for detecting tissue of origin in liquid biopsy. Conventional NGS methods can have limited utility in fragmentomic analyses, as NGS tends to output reads shorter than 500 base pairs [220,221]. Some recent approaches for obtaining fragmentomic signatures using NGS include linked-read sequencing used for obtaining long-read information from short reads [222], and the DNA evaluation of fragments for early interception (DELFI) approach [223,224]. Another form of NGS, called hybrid capture sequencing, has been used to measure the ctDNA fragment length and detect minimal residual disease in stage II-IIIA NSCLC patients [225].

Long-read sequencing technologies, including Oxford Nanopore Technologies (ONT) and PacBio devices, can directly read long nucleic acid fragments and their methylation patterns [226,227,228]. This makes them well suited to fragmentomics studies and the analysis of repetitive regions of the genome. Recently, a customized workflow using ONT was developed to sequence ctDNA. This successfully detected copy number alterations in lung cancer patients [229]. PacBio technology has recently been used to assess the length and methylation scores of plasma DNA in hepatocellular carcinoma (HCC) patients. An HCC methylation score was developed to differentiate liver-derived ctDNA from other tissue DNA. Using longer DNA fragments improved the diagnostic capability of the HCC score [230]. 

### 5.4. Single-Cell Sequencing

The motivation to perform single-cell sequencing of CTCs is to obtain an up-to-date molecular profile of metastatic cancer cells, to study mechanisms of metastasis, and to characterize the cells’ progression and evolution. The low volume of DNA or RNA derived from a CTC necessitates whole-genome sequencing (WGS) or whole-transcriptome amplification (WTA) to meet the required limit of detection for sequencing [231]. A recent comparison of four WTA methods for use in the single-cell sequencing of CTCs found that multiple annealing and looping-based amplification cycles (MALBAC) followed by low-pass WGS outperformed the other three methods in coverage breadth, reproducibility, and uniformity, though none of the methods met the sensitivity or specificity required for clinical use [232]. A study using MALBAC for the single-cell sequencing of CTCs isolated from lung cancer patients found that while most genomic features were heterogeneous among all CTCs, copy number variations were consistent for CTCs derived from the same patient and could be used to discriminate between lung adenocarcinoma and SCLC [233]. Another study that performed single-cell sequencing of CTCs derived from SCLC patients observed that copy number alterations were predictive of the response to chemotherapy and that the evolutionary history of cells could be deduced from copy number alterations detected at different times over the course of treatment [234]. Facilitated by WGS and WTA methods, the single-cell sequencing of CTCs is a promising approach to monitoring the progression of metastatic lung cancer.

## 6. Emerging Approaches for Liquid Biopsy of Lung Cancer

Nucleic-acid-based methods to detect ctDNA and CTC or EV-derived biomarkers in liquid biopsy are well studied and used with increasing frequency in clinical settings. Here, we highlight two less common approaches to biomarker detection that show promise for NSCLC monitoring: the lung microbiome and tumor-educated platelets (TEPs).

### 6.1. Microbiome

The interest in understanding the link between the microbiome and the development and progression of lung cancer has increased in recent years. It is known that the microbiome plays an important role in human health and disease by modulating a host’s innate and adaptive immune system, immune responses, and metabolism, and by protecting from invading pathogens [235,236,237]. The tumor microenvironment (TME) influences cancer progression and therapy responses [238] and many recent studies have demonstrated associations between the TME and microbiota composition and responses to immune checkpoint inhibitors in cancer patients, including NSCLC patients, suggesting that modulation of the microbiota through diet, probiotics, and fecal microbiota transplantations (FMT) could improve treatment efficacy [238,239,240,241,242,243,244]. There is also a possibility that the lung microbiome change could be used as a biomarker for detecting clinical phenotypes [245]. Some studies have focused on the analysis of the tumor tissue microbiome and showed evidence of its link with metastasis [246,247], while others have focused on liquid biopsies and the analysis of the microbiome as potential markers. Epithelial brushing samples collected via bronchoscopy from patients with lung cancer, incident cancer, and those who did not develop cancer after a 10-year follow-up were analyzed through 16S ribosomal RNA gene (rDNA) sequencing. Results suggested that a shift in *Veillonella*, *Streptococcus*, *Prevotella*, and *Paenibacillus* may occur in the airways of patients with incident lung cancer months before the bronchoscopy date [245]. Emerging evidence has demonstrated that alterations of circulating microbiome DNA in blood could serve as promising non-invasive biomarkers for cancer detection, including lung cancer [236]. Microbe-derived plasma cfRNAs were consistently detected by different computational methods across lung cancer and four other cancers, and the highest bacterial abundance was found for *Proteobacteria*, followed by *Firmicutes* and *Actinobacteria* [248]. NGS analysis of bronchoalveolar lavage fluid showed that the bacterial diversity was lower in lung cancer than in benign lung nodules, where four species of *Porphyromonas somerae*, *Corynebacterium accolens*, *Burkholderia cenocepacia*, and *Streptococcus mitis* were enriched in lung cancer compared with benign lung nodules [249]. The same study also concluded that the abundance of the lung microbiota is closely related to the development of infiltrating adenocarcinoma. Moreover, 16S rRNA gene sequencing on malignant pleural effusion (MPE) samples and controls has shown that there are compositional differences among pleural effusions related to non-malignant, para-malignant, and malignant disease [250]. The pleural fluid of MPE-Lung and Mesothelioma was associated with *Rickettsiella*, *Ruminococcus*, *Enterococcus*, and *Lactobacillales*. Mortality in MPE-Lung was associated with enrichment in *Methylobacterium*, *Blattabacterium*, and *Deinococcus*. 

These findings suggest the potential use of lung microbiome profiling using liquid biopsies as a promising method of diagnosis. However, more research is needed with robust cohorts of patients to understand the relationship between the microbiome and lung cancer to validate findings.

### 6.2. Tumor-Educated Platelets (TEPs)

Besides CTCs, other types of circulating cells, such as platelets and circulating immune cells, can have altered abundances or phenotypes due to cancer, and can also be investigated as biomarkers [251,252]. Platelets and macrophages are known to be exploited by cancer cells to support the immune evasion of tumor cells and angiogenesis. Platelets in particular are abundant in circulation, easy to isolate, and known to be altered in cancerous patients, so they present an attractive target for liquid biopsy. TEPs are defined as platelets derived from cancer patients that have distinctively altered RNA and protein content [253]. Potential mechanisms underlying the “education” of platelets include the absorption of factors derived from tumors, cell-signal-driven changes in platelets’ RNA processing, and modified platelet manufacturing by megakaryocytes. Tumor-educated platelets may influence the overexpression of additional cancer-promoting signals, including genes associated with EMT, such as TGFβ, and invasion-promoting genes such as matrix MMP-9 [254]. Metastatic NSCLC has been shown to be associated with the significant downregulation of several genes in platelets [255], and mRNA signatures have been used to detect NSCLC using TEPs [256]. A diagnostic sequencing pipeline called ThromboSeq has been able to identify characteristic changes in RNA splicing in TEPs and has been used to distinguish both early- and late-stage NSCLC patients from healthy controls [257,258]. Given the relatively high accuracy of TEP RNA-seq panels, TEPs are a promising emerging biomarker for lung cancer.

## 7. Conclusions

Liquid biopsy is a minimally invasive technique that has shown promise in detecting tumor biomarkers in the body fluids of patients with lung cancer. Due to its advantages, such as test repeatability and suitability for patients with poor clinical conditions, it offers an option for patients who are not suitable candidates for traditional solid tissue biopsies. Monitoring and identifying minimal residual disease in patients undergoing treatment is of great importance in clinical practice as it enables early therapeutic intervention with the potential of reducing disease recurrence.

Furthermore, significant advancements have been observed in the ability of liquid biopsies to identify specific mutations associated with lung cancer in advanced disease stages, which can facilitate individualized treatment strategies and lead to enhanced treatment responses and survival rates. However, continued research is required to improve the sensitivity and specificity of liquid biopsy tests, with the ultimate goal of enabling the detection of serious clinical complications in lung cancer, such as recurrence and distant metastases, before they become clinically evident. With ongoing technological advancements, liquid biopsy tests will become a routine part of clinical practice for lung cancer patients, providing personalized therapies and improving patient outcomes.

## Figures and Tables

**Figure 1 ijms-24-08894-f001:**
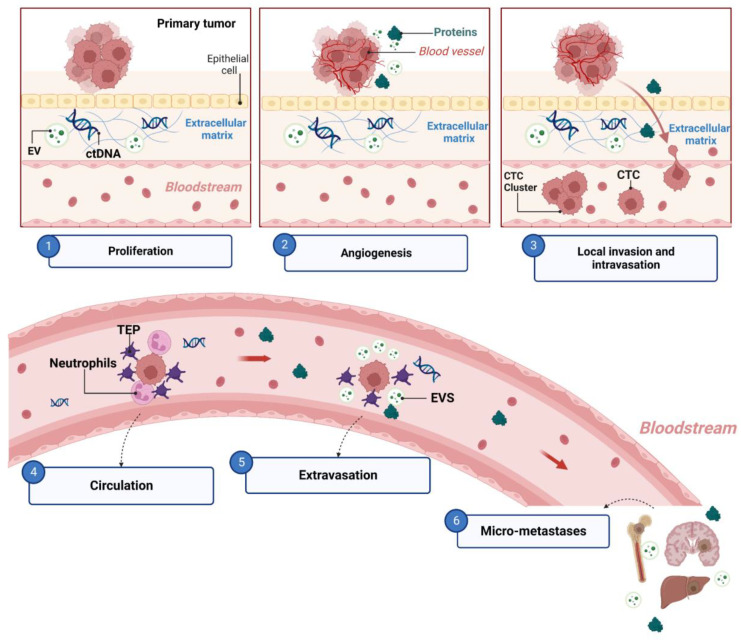
The metastatic cascade and associated biomarkers. (1) During cancer cell proliferation, DNA and extracellular vesicles (EVs) may be released from necrotic or apoptotic cells. (2) Tumor-derived EVs promote tumor angiogenesis. Proteins such as matrix metalloproteinases (MMPs) and vascular endothelial growth factor (VEGF) contribute to angiogenesis by mechanisms such as degrading basement membranes and other extracellular matrix (ECM) components, allowing endothelial cells to detach and migrate into new tissue, and by releasing ECM-bound proangiogenic factors (bFGF, VEGF, and TGFβ). (3) EVs released by cancer cells contribute to endothelial-to-mesenchymal transition. MMPs and VEGF promote intravasation. (4) Platelets and neutrophils can aid circulating tumor cells (CTCs) in evading the immune system and surviving in the bloodstream by forming clusters or aggregates with CTCs. (5) Tumor-educated platelets (TEP) can facilitate the extravasation of CTCs by releasing EVs that can modulate the dynamics of endothelial cell junctions and increase the permeability of endothelial cells. (6) Tumor-derived exosomes contribute to the formation of the pre-metastatic niche and influence the site of metastatic colonization. VEGF is involved in the formation of pre-metastatic niches through direct and indirect mechanisms. Tumor- or host-cell-derived VEGF assists disseminated tumor cell colonization by neovascularization and generating a suppressive immune microenvironment. Moreover, in some cases, VEGF triggers the phenotypic switch from dormancy to proliferative status and restarts the colonization process.

**Table 1 ijms-24-08894-t001:** Clinical utility of liquid biopsy biomarkers in cancer.

Body Fluid	Biomarkers Detected	Examples of Applications
Plasma	ctDNA to determine resistance mechanisms in patients with advanced NSCLC [50] Prognostic biomarkers: miR-10b-5p, miR-23b-3p, and miR-21-5p [51] Gene mutations including *ALK*, *EGFR*, *KRAS* with droplet digital PCR [52]	Plasma protein signatures reflect tumor biology [53] Increased RP11-438N5.3 lncRNA levels for patients’ prognosis [54] *EGFR*-status-related miRNA panel derived from exosomes isolated from metastatic disease [55]
Serum	Levels of metabolites in disease progression: lactic acid, benzoic acid, and fumaric acid [56] Prolactin, an early preventive factor in metastatic NSCLC for poor clinical outcomes [57] High exosomal microRNAs in advanced disease and poor survival: miR-378 and miR-214 [58]	CA125 levels are an indicator of metastasis to the liver [59] CYFRA 21-1 can be correlated with worsened PFS and OS in metastatic patients [60]
Sputum	Chromosomal aneusomy by FISH predicts lung cancer incidence [61] Sputum supernatant has comparable mutation profiling to plasma samples of NSCLC, with advantage of convenience in collection [62]	Use of nomograms to demonstrate the utility of sputum samples for genome profiling [63] Sputum as an alternative source for somatic variation profiling with NGS [64]
BAL	Cytologic examination of BAL has a comparable diagnostic yield to other endoscopic techniques for metastasis [65]	BAL better reflect the cancer proteome than serum samples [66]
Pleural effusion	MMP-9, cathepsin-B, C-reactive protein, chondroitin sulfate marker panel, and CA19-9, CA15-3, kallikrein-12 panel, are highly discriminative for malignant vs. tuberculous effusion, or lung adenocarcinoma vs. mesothelioma [67,68]	Detection of extracellular vesicles and cfDNA in pleural effusions enhances *EGFR* genotyping of adenocarcinoma patients [69]
Urine	*EGFR* mutations [70]	ctDNA *EGFR* mutation testing detects T790M mutations overlooked in tissue biopsies due to sample quality or tumor heterogeneity [71] ddPCR and Illumina MiSeq to monitor *EGFR* alterations during treatment [72]
CSF	*EGFR*, *ROS1*, *ALK*, *BRAF*, and/or *EGFR* T790M mutations [35]	Profiling for actionable mutation rate (*EGFR*, *ROS1*, *ALK*, *BRAF*) and resistance mutation rates (*EGFR* T790M mutation) [35] Higher detection sensitivity for leptomeningeal metastasis by CTCs than with MRI [72]

Abbreviations: cfDNA: cell-free DNA; ctDNA: circulating tumor DNA; CTCs: circulating tumor cells; NSCLC: non-small-cell lung cancer; NGS: next-generation sequencing; lncRNA: long non-coding RNA; MS: mass spectrometry; BAL: bronchoalveolar fluid; CSF: cerebrospinal fluid; ddPCR: droplet digital PCR; FISH: fluorescence in situ hybridization; PFS: progression-free survival; OS: overall survival; MRI: magnetic resonance imaging.

## Data Availability

Not applicable.

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
