# Peer review of "Liquid Biopsy in Lung Cancer: Biomarkers for the Management of Recurrence and Metastasis"

_ijms, 2023, doi:10.3390/ijms24108894_

Round 1

Reviewer 1 Report

The manuscript focuses on a systemic analysis of literature data about the identification of jet established and promising role of liquid biopsy in clinical stage of lung cancer patients. Accordingly, the manuscript is well structured and overview all the most clinical topics related to liquid biopsy specimens. In my opinion, few minor considerations should be approached to improve the readibility of the manusscript.

- In the manuscript, please, could the authors review the technical approaches section distinguishing between RT-PCR and sequencing based methods? As regards, I would suggest to cosider NCounter as a specific approach not included in previously mentioned cathegories. In addition, I would also reccomend to add a table where technical approaches are listed in terms of sensitivity, multiplexing, TAT and LOD.

No other comments

Reviewer 2 Report

In general, the review paper is well-structured and easy to follow. The manuscript aim to summarize available and novel approaches to liquid biopsy tests for lung cancer metastases and recurrence detection, and also describe their applications in clinical practice.

Minor concerns:

1. It is highly recommended the authors added a paragraph to describe clinical utility of liquid biopsy for detecting metastasis of lung cancer (Table 1) in order to make it easier to read.

2. The application of liquid biopsy in predicting the outcomes of immunotherapy in lung cancer should be added in the text.

3. There are multiple studies that have analyzed this topic (PMID: 37002211, PMID: 36797152, PMID: 37057121, PMID: 36384005, PMID: 37054576, and etc) so it lacks novelty. What does your manuscript add?

Round 2

Reviewer 2 Report

The authors have addressed all of the concerns.